# Anti PD-1/Anti PDL-1 Inhibitors in Advanced Gastroesophageal Cancers: A Systematic Review and Meta-Analysis of Phase 2/3 Randomized Controlled Trials

**DOI:** 10.3390/pharmaceutics14091953

**Published:** 2022-09-15

**Authors:** Kanak Parmar, Sai Subramanyam, Kristopher Attwood, Duke Appiah, Christos Fountzilas, Sarbajit Mukherjee

**Affiliations:** 1Department of Internal Medicine, Texas Tech University Health Sciences Center, Lubbock, TX 79430, USA; 2Department of Internal Medicine, Appalachian Regional Healthcare, Harlan, KY 40831, USA; 3Department of Epidemiology, Roswell Park Comprehensive Cancer Center, Buffalo, NY 14263, USA; 4Department of Public Health, Texas Tech University Health Sciences Center, Lubbock, TX 79430, USA; 5Department of Medicine, Roswell Park Comprehensive Cancer Center, Buffalo, NY 14263, USA

**Keywords:** gastric cancer, esophageal cancer, gastroesophageal junction cancer, immunotherapy

## Abstract

**Importance:** Immune checkpoint inhibitors (ICI) have revolutionized the treatment for gastroesophageal cancers (GEC). It is important to investigate the factors that influence the response to anti-PD-1/PD-L1 ICIs. **Objective:** To assess the benefits of PD-1/PD-L1 ICIs in advanced GEC and perform subgroup analysis to identify patient populations who would benefit from ICI. **Data sources:** PubMed, Embase, Scopus, and the Cochrane Library databases were systematically searched from database inception to September 2021 for all relevant articles. We also reviewed abstracts and presentations from all major conference proceedings including relevant meetings of the American Society of Clinical Oncology (ASCO), and the European Society for Medical Oncology (ESMO) during the last four years (2018 to 2021) and reviewed citation lists. **Study selection, data extraction, and synthesis:** Full articles and presentations were further assessed if the information suggested that the study was a phase 2/3 randomized controlled trial (RCT) comparing PD-1/PD-L1 inhibitor either alone, or in combination with standard therapy vs. standard therapy in advanced GEC. The full text of the resulting studies/presentations and extracted data were reviewed independently according to PRISMA guidelines. **Main outcomes and measures:** The main outcomes were OS, PFS, and treatment-related adverse events (TRAEs). **Results:** A total of 168 studies were assessed for eligibility, and 17 RCTs with 12,312 patients met the inclusion criteria. There was an OS benefit in the overall population with ICIs (HR 0.78; 95% CI 0.73–0.83 *p* < 0.001). Immunotherapy showed better OS benefit in males (HR 0.77 95% CI 0.72–0.83; *p* < 0.001) than females (HR 0.89; 95% CI 0.80–0.99 *p* < 0.03), esophageal primary tumors (HR 0.70 95% CI 0.64–0.76 *p* < 0.001) vs. gastric cancer (HR 0.84 95% CI 0.74–0.94 *p* 0.002) or GEJ cancer (HR 0.84 95% CI 0.72–0.98 *p* 0.024) and in squamous cell carcinoma (HR 0.71 95% CI 0.66–0.77 *p* < 0.001) vs. adenocarcinoma (HR 0.85 95% CI 0.78–0.93 *p* < 0.001). PD-L1 positive patients seemed to benefit more (HR 0.74 95% CI 0.67–0.82 *p* < 0.001) compared to PD-L1 negative patients (HR 0.86 95% CI 0.74–1.00 *p* < 0.043), and Asians showed OS benefit (HR 0.76 95% CI 0.67–0.87 *p* < 0.001) compared to their White counterparts (HR 0.92 95% CI 0.74–1.14; *p* 0.424). **Conclusions and relevance:** ICIs improve survival in advanced GEC without significantly increasing the side effects. However, certain subgroups of patients such as males, Asians, and those with esophageal primary, PD-L1 positive tumors and squamous cell carcinoma benefit more from such treatments. Further translational research is needed to understand the mechanistic links and develop new biomarkers.

## 1. Introduction

Gastroesophageal cancers (GEC) consist of the esophageal, gastroesophageal junction (GEJ), and gastric cancers and are the most prevalent gastrointestinal cancers worldwide. More than 600,000 new esophageal cancer cases are detected annually, claiming 544,000 lives [1]. On the other hand, more than one million new gastric cancer cases are detected annually worldwide, leading to an estimated 769,000 deaths [1].

A combination regimen with platinum plus fluoropyrimidine has long been the preferred first-line palliative treatment for advanced GEC [2,3]. However, the prognosis remains poor [4]. Recently, first-line standard-of-care treatment in advanced GEC has changed, given the positive randomized phase 3 trials combining ICI with first-line chemotherapy. CheckMate 649 led to the approval for nivolumab, an anti-PD-1 ICI, in combination with first-line chemotherapy in gastroesophageal adenocarcinoma, irrespective of the PD-L1 status [5]. In PD-L1 negative/ low-expressing gastroesophageal cancers, benefit of ICI is still a matter of debate and nivolumab’s approval in Europe is still limited to PD-L1 ≥ 5. The approval of pembrolizumab, another anti-PD-1 ICI, was based on two positive trials, KEYNOTE 590 in esophageal cancer and KEYNOTE 811 in HER-2 positive gastric cancer [6,7]. In the KEYNOTE 590 study, pembrolizumab plus chemotherapy was superior to placebo plus chemotherapy for OS in patients with esophageal squamous cell carcinoma (SCC) and PD-L1 CPS of 10 or more (median 13.9 months vs. 8.8 months; HR 0.57 95% CI 0.43–0.75; *p* < 0.0001) and in all randomized patients (12.4 months vs. 9.8 months; HR 0.73 95% CI 0.62–0.86 *p* < 0.0001). Preliminary results of KEYNOTE 811 for patients with HER2+ metastatic gastric or gastroesophageal junction (G/GEJ) cancer comparing pembrolizumab, trastuzumab and chemotherapy vs. investigator choice chemotherapy showed remarkable improvement in objective response rate (ORR) (74.4% vs. 51.9%) and complete response (CR) rate (11.3% vs. 3.1%). In terms of side effects, the ICIs are known to have a manageable toxicity profile [8]. But even though the added toxicity from ICI is limited, the financial burden can be substantial and a consensus on clinical utility of ICIs has not been reached. Therefore, we sought to investigate if there are any clinical factors associated with the benefit from ICIs in advanced GEC.

## 2. Summary

### 2.1. Question

What is the role of anti-programmed death-1 (anti-PD-1)/ programmed death ligand-1 (PD-L1) therapy in advanced gastroesophageal cancers (GEC), and which sub-population benefit from this therapy?

### 2.2. Findings

In this meta-analysis of 17 relevant phase 2/3 randomized controlled trials (RCTs) that used anti-PD-1/PD-L1 inhibitor treatment, there was an overall survival (OS) benefit for using such treatment in the whole population. On subgroup analysis, the OS benefit was more pronounced in Asians vs. Whites, males vs. females, squamous vs. adenocarcinoma, esophageal primary vs. gastric and gastroesophageal junction (GEJ), and PD-L1 positive vs. PD-L1 negative tumors.

### 2.3. Meaning

Anti-PD-1/PD-L1 inhibitors can provide benefit in patients with advanced GEC that appears to be driven by OS benefit in patients with squamous histology, esophageal primary, and PD-L1 positive tumors. There is an unmet need for developing additional biomarkers.

## 3. Methods

### 3.1. Data Sources and Searches

The present systematic review and meta-analysis were performed according to the Preferred Reporting Items for Systematic Reviews and Meta-Analyses (PRISMA) Statement and the guideline for meta-analysis of observational studies in epidemiology (MOOSE) [9,10]. PubMed, Embase, Scopus, and the Cochrane Library were searched for all relevant articles from database inception to September 2021. We also reviewed abstracts and presentations from all major conference proceedings from the American Society of Clinical Oncology (ASCO), and the European Society for Medical Oncology (ESMO) during the last four years (2018 to 2021) and reviewed citation lists. The following search algorithm was used: “gastric cancer” OR “gastroesophageal cancer” OR “esophagus cancer” OR “gastric adenocarcinoma” OR “esophageal adenocarcinoma” OR “esophageal squamous cell carcinoma” OR “gastric squamous cell carcinoma” AND “PD-1 inhibitor” OR “pdl-1 inhibitor” OR “anti-pdl-1” OR “nivolumab” OR nivo OR pembrolizumab OR pembro OR atezolizumab OR avelumab OR durvalumab OR “immune checkpoint inhibitor” OR “imfinzi” OR “bavencio” OR tecentriq OR “keytruda” OR “opdivo”. Both randomized phase 2 and phase 3 trials were allowed. The analysis was registered on Prospero under registration code CRD42022308323.

### 3.2. Selection Criteria

Articles were included if RCTs compared the OS of patients with advanced GEC treated with PD-1/PDL-1 inhibitors vs. standard care, regardless of the therapeutic line. The search was restricted to the English language only. Title and abstracts of all articles retrieved using the search strategy were initially screened, reviewed, and verified independently by two authors (KP and SS), with disagreements mediated through discussion with a third review author (SM). Studies with unreported survival data were not included in this analysis (Figure 1).

### 3.3. Data Extraction

The full texts of potentially eligible articles/presentations were reviewed by SS and KP, with disagreements mediated by SM. Two authors (KP and SS) used a single pre-formatted extraction form for all included articles to extract the data. A discussion was made with a third author (SM) to resolve any discrepancies. Prespecified subgroups included patient’s age at the time of randomization (categorized as age ≤65 years vs. ≥65 years), gender (female vs. male), ethnicity (Asians vs. Whites), Eastern Cooperative Oncology Group performance status (ECOG PS) (0 vs. 1), primary tumor location (esophageal vs. gastric vs. GEJ), and histological subtype (squamous vs. adenocarcinoma). The Cochrane Risk of Bias Tool was used to assess the risk of bias for each RCT [11]. The quality of the studies was assessed independently by our two reviewers (KP and SS), and disagreements were mediated by consensus with a third author (SM).

### 3.4. Statistical Methods

A meta-analysis was conducted on the survival outcomes (OS and PFS), where hazard ratios (HR) were obtained, with 95% confidence intervals, using the standard random-effects model. The same modeling approach was used for treatment-related adverse events (TRAEs) where odds ratios were estimated. Statistical heterogeneity in the results of the trials was assessed by the χ^2^ test and was expressed by the I^2^ index, as described by Higgins and colleagues. The I2 statistic represented the variability in the meta-analysis attributed to study heterogeneity. These analyses were applied to the overall cohort of studies and within the specific demographic/clinical sub-cohorts. All analyses were conducted in SAS v9.4 (Cary, NC, USA) at a significance level of 0.05. Publication bias was evaluated with the Egger test [12].

## 4. Results

### 4.1. Study Selection

A total of 5319 studies were screened from electronic databases out of which 56 studies were assessed for eligibility by reviewing full-text articles (Figure 1). Only 17 RCTs with 12,312 patients met the inclusion criteria. The included study characteristics are shown in Table 1 and Table 2. Nine RCTs (*n* = 6625) compared anti-PD-1/PD-L1 ICIs plus chemotherapy with chemotherapy alone [5,6,13,14,15,16,17,18,19]. Nine RCTs (*n* = 4989) compared anti-PD-1/PD-L1 ICIs with chemotherapy [5,19,20,21,22,23,24,25]. Lastly, two RCTs (*n* = 698) compared anti-PD-1/PD-L1 ICIs with placebo [26,27]. There were 10 studies in the first line setting, 5 studies in the second line setting, and 2 studies in the third or higher line setting.

### 4.2. Overall Study Population

In the overall population, ICIs significantly improved OS (HR 0.78 95% CI 0.73–0.83 *p* < 0.001). There was no statistically significant improvement in PFS (HR 0.86 95% CI 0.71–1.03 *p* < 0.101) (Figure 2A,B). Anti PD-1/PD-L1 showed statistically significant OS benefit in both first (HR 0.79; 95% CI 0.73–0.85 *p* < 0.001) and second line (HR 0.74 95% CI 0.66, 0.81 *p* < 0.001) settings. No OS benefit was observed in the third-line setting (HR 0.82 95% CI 0.47–1.44 *p* < 0.496). The improvement in OS was significant across histology but appeared more pronounced in squamous cell carcinoma (HR 0.71 95% CI 0.66–0.77 *p* < 0.001) compared to adenocarcinoma (HR 0.85 95% CI 0.78–0.93; *p* < 0.001) (Figure 3A,B). Furthermore, males appeared to benefit more compared to females (HR 0.77 95% CI 0.72–0.83 *p* < 0.001 vs. HR 0.89 95% CI 0.80–0.99 *p* < 0.033). Benefit was present across all age groups: younger (HR 0.80 95% CI 0.74–0.86 *p* < 0.001) and older age groups (HR 0.79 95% CI 0.72–0.87 *p* < 0.001) (Appendix A). The OS improvement remained significant across different anatomic locations though more pronounced for esophageal primary tumors (esophageal cancer: HR 0.70 95% CI 0.64–0.76 *p* < 0.001 vs. gastric cancer: HR 0.84 95% CI 0.74–0.94 *p* 0.002 vs. GEJ cancer: HR 0.84 95% CI 0.72–0.98 *p* 0.024). Performance status did not seem to affect OS with ECOG 0 (HR 0.78 95% CI 0.69–0.89 *p* < 0.001) and ECOG 1 (HR 0.77; 0.70–0.84 *p* < 0.001). The OS benefit was statistically significant in patients with PD-L1 positive but not PD-L1 negative tumors (HR 0.74 95% CI 0.67–0.82 *p* < 0.001 vs. HR 0.86 95% CI 0.74–1.00 *p* 0.043 respectively). Patients of Asian descent had significant improvement in OS (HR 0.76 95% CI 0.67–0.87 *p* < 0.001) whereas Whites did not seem to benefit from anti PD-1/PD-L1 (HR 0.92 95% CI 0.74–1.14 *p* 0.424). There was no difference in TRAEs, grade 3+ AE or any AE with ICI as compared to chemotherapy (HR 1.27 95% CI 0.62–2.58 *p* 0.511; HR 1.03 95% CI 0.59 1.77 *p* 0.93; HR 0.78 0.35 1.72 *p* 0.54).

### 4.3. Anti PD-1/PD-L1 plus Chemotherapy vs. Chemotherapy

Anti-PD-1/PD-L1 in combination with chemotherapy lead to superior OS compared to chemotherapy alone (HR 0.74 95% CI 0.68–0.81 *p* < 0.001). PFS was also significantly improved with incorporation of anti-PD-1/PD-L1 agents (HR 0.64 95% CI 0.59–0.70 *p* < 0.01) (Figure 4A,B). Again, the benefit was more pronounced for patients with esophageal primary (HR 0.66; 95% CI 0.59–0.75 *p* < 0.001) and gastric cancer (HR 0.76 95% CI 0.68, 0.84 *p* < 0.001) vs. GEJ cancers (HR 0.91 95% CI 0.74–1.11 *p* 0.351). OS benefit was consistent amongst different age groups (Appendix A). Significant improvements in OS were noted in both Asian (HR 0.73 95% CI 0.57–0.92 *p* 0.008) and Whites (HR 0.80 95% CI 0.71–0.90 *p* < 0.001) patients (Appendix A). While OS benefit was significant for both genders, males appear to benefit more than females (HR 0.73 95% CI 0.68–0.79 *p* < 0.001 vs. HR 0.84 95% CI 0.72–0.97 *p* 0.02) (Appendix A). Significant improvements were observed in both squamous cell carcinoma (HR 0.68 95% CI 0.61, 0.75 *p* < 0.001) and adenocarcinoma (HR 0.80 95% CI 0.73–0.88 *p* < 0.001). Use of ICI plus chemotherapy in patients with PD-L1-positive tumors had more pronounced OS effect (HR 0.69 95% CI 0.58–0.81 *p* < 0.001) compared to PD-L1 negative tumors (HR 0.84 95% CI 0.75–0.94 *p* < 0.002). Anti PD- 1/PD-L1 added to chemotherapy increased the risk of TRAEs, grade 3+ AE (HR 5.96 95% CI 2.60–13.68 *p* < 0.001; HR 3.40 95% CI 1.83–6.31 *p* < 0.001).

### 4.4. Anti-PD-1/PDL-1 vs. Chemotherapy

When used as monotherapy, ICIs improved OS compared to chemotherapy (HR 0.84 95% CI 0.76–0.92 *p* < 0.001) without significant effect on the PFS (Appendix A). Anti-PD-1/PD-L1 significantly improved OS in patients with esophageal primary (HR 0.74 95% CI 0.66–0.82 *p* < 0.001), Asian origin (HR 0.83 95% CI 0.70–0.99 *p* 0.037), males (HR 0.87 95% CI 0.79–0.96 *p* 0.004), PD-L1 positive (HR 0.81 95% CI 0.71–0.93 *p* 0.002), and squamous cell histology (HR 0.75 95% CI 0.67–0.84 *p* < 0.001) (Appendix A). The risk of any AE, TRAEs, grade 3+ adverse events was lower in patients treated with ICI compared to chemotherapy (HR 0.44 95% CI 0.23–0.86 *p* 0.016; HR 0.30 95% CI 0.17–0.53 *p* < 0.001; HR 0.28 95% CI 0.17–0.46 *p* < 0.001). Anti PD-1/PD-L1 showed statistically significant OS benefit in both first (HR 0.88; 95% CI 0.80–0.98 *p* < 0.02) and second line (HR 0.75 95% CI 0.67–0.83 *p* < 0.001) settings. Only one study was included in the third-line setting with no OS benefit (HR 1.10 95% CI 0.86–1.40 *p* 0.439) (Appendix A).

### 4.5. Anti PD-1/PDL-1 vs. Placebo

When comparing anti-PD-1/PDL-1 vs. placebo, ICI had no OS benefit (HR 0.74 95% CI 0.50–1.09; *p* < 0.125). PFS results included only one study with benefit from ICI (HR 0.60 95% CI 0.48–0.75; *p* < 0.001).

### 4.6. Publication Bias

The results of Egger’s tests indicate no statistically significant biases (Appendix A). When we look at OS, it is a perfect funnel plot (*p* 0.66). PFS has a symmetric funnel plot, but we see that several studies (with high precision) demonstrate benefit or lack of benefit for immunotherapy (*p* 0.79). The funnel plot is symmetric for TRAE and behaves similar to PFS (*p* 0.14). For any AE, there really are fewer studies to make a formal assessment (*p* 0.31). For the grade 3+AEs, there does appear to be some potential bias, where studies with higher precision (i.e., lower standard error) show a benefit for immunotherapy, but studies with lower precision demonstrate the opposite (*p* 0.84). Overall, publication bias may not be a significant concern.

## 5. Discussion

The benefit of chemo-immunotherapy in the frontline metastatic GEC was shown in the landmark trials: KEYNOTE 590, CheckMate 649, CheckMate 648, and KEYNOTE 811 [5,6,19,28]. Based on the OS benefits in KEYNOTE-590 and CHECKMATE 649 studies, ICIs are now routinely used in the frontline treatment of metastatic human epidermal growth factor receptor-2 (HER 2) negative GEC. KEYNOTE-811 is an ongoing randomized, double-blind, phase 3 trial comparing pembrolizumab combined with trastuzumab and chemotherapy with trastuzumab and chemotherapy in HER-2 positive gastric and GEJ cancers [28]. The initial results showed a statistically significant increase in objective response rate (ORR) and duration of response, which led to Food and Drug Administration (FDA) approval of pembrolizumab in this setting. The OS data are not yet mature, and therefore, it was excluded from our analysis. On the other hand, there have been several negative chemo-immunotherapy studies in advanced GEC [23,26,27]. Moreover, further subgroup analysis of KEYNOTE 590 and CHECKMATE 649 studies showed that certain subgroups such as PD-L1 low or negative patients do not derive as much benefit from the use of ICIs [5,6]. This currently presents a clinical dilemma before clinicians for selecting the proper patient population for ICIs. Therefore, the meta-analysis was performed to identify the population where ICIs add most benefit. Our analysis showed that when added to chemotherapy, ICIs improve OS compared to chemotherapy alone. In the immunotherapy vs. chemotherapy setting, there was an OS but no PFS benefit. This is contrary to the previous meta-analyses by Kundell and Wang, which included five and three RCT’s respectively, and showed no benefit in OS with ICI [29,30].

Our data showed that ICIs seem to benefit males more than females on subgroup analysis. This benefit was persistent when ICI was combined with chemotherapy or immunotherapy was given alone. It is known that the regulation of the immune system is influenced by sex hormones, including estrogen, progesterone, and androgens [31]. Multiple studies in various cancer databases showed that at the molecular level, there were more tumor mutational burden (TMB) and more cancer germline antigens in males, which can be aberrantly expressed in a variety of human malignancies [32,33]. This provides more targets for ICIs and may explain a better response [33]. Recent studies have also shown that women may have an increased risk of immune-related AE than men, especially endocrinopathies and pneumonitis [31]. Thus, gender plays a vital role in ICI response, and strategies tailored to the patient’s gender may improve outcomes. Further translational studies can provide more insight into the effect of gender in ICIs.

Another essential subgroup for our analysis was age. GEC is predominantly a disease of older age. This subgroup of patients is exposed to a significant risk factor for cancer cachexia, which have been linked to poor prognosis [34]. However, our data showed that immunotherapy benefits are consistent across all age groups corroborating the previously reported meta-analysis [34]. Various studies have highlighted that patient aged 80 years or older comprise only 4% of cancer clinical trials [35,36,37]. A study by Athauda et al. showed similar OS between younger and older patients in GEC on chemotherapy [38]. They also showed comparable rates of any-grade and severe immunotherapy-related AEs (G3-G4) across geriatric age subgroups [38]. Thus, based on our study older age group should not be excluded from trials testing ICI because old patients respond equally well to ICI therapy.

Biomarker discovery and validation is an area of great interest in GEC to identify tumors that are more likely to benefit from ICI [39]. The PD-L1 combined positive score (CPS) is the most common method of assessing PD-L1 expression in GEC. The tumor proportion score (TPS) score is the ratio of the number of PD-L1–expressing tumor cells to all tumor cells. The combined positive score (CPS) was developed to consider the expression of PD-L1 on tumor and immune cells combined. CPS is the ratio of all PD-L1–expressing cells (tumor cells, lymphocytes, macrophages) to the number of all tumor cells. In gastric cancer, CPS has been found to be a better biomarker than TPS [40]. Measurement of PD-L1 expression with the CPS in patients in KEYNOTE-059 trial showed that the objective response to pembrolizumab was significantly associated with CPS but not with TPS [41]. However, there is a discrepancy in the assessment methods as they require different staining protocols, equipment, and cut-offs, and standardized methods and definitions of PDL-1 positivity are required. In a recent study, Yeong et al. showed that Dako 28–8 assay results in higher PD-L1 CPS positives compared to the Dako 22C3 assay [42]. Therefore, these assays should not be used interchangeably while assessing the PD-L1 CPS scores in patients.

A recent meta-analysis showed that esophageal squamous carcinoma patients benefit from ICIs, however, the efficacy is dependent on PD-L1 CPS status, using a cut-off of 10 [43]. In our analysis, PD-L1 positive patients were defined as CPS PD-L1 score ≥1 and they seemed to benefit from ICIs, unlike PD-L1 negative patients. In another study by Zhou et al., subgroup analysis of CheckMate-649 study showed that the ICI-chemotherapy combination do not lead to better PFS and OS in PD-L1 CPS 1–4. Similarly, no benefit was found with the addition of pembrolizumab in the PD-L1 CPS 1–9 group in KEYNOTE-062 study [44]. These data, taken together, argue against the widespread use of ICI in gastroesophageal adenocarcinoma; and risks and benefits should be thoroughly considered before using ICIs in low PD-L1 expressing patients. Nevertheless, it should be kept in mind that PD-L1 expression shows spatial and temporal heterogeneity and PD-L1 CPS score is subject to inter-observer variation even while using the same assay. Therefore, additional biomarkers are needed to define the complex immune microenvironment and predict response from ICI therapy.

Several predictive biomarkers have been identified in addition to PD-1/PD-L1 expression including MSI, tumor mutational burden (TMB), and Epstein Barr virus (EBV) [45]. A post-hoc analysis of the KEYNOTE-059, KEYNOTE-061, and KEYNOTE-062 showed that the MSI-H gastric/GEJ cancer patients derived benefit from Pembrolizumab, regardless of the line of therapy in which it was received [46]. A recent study showed that nearly 9% of GC patients are Epstein-Barr Virus (EBV) positive, and EBV-associated gastric cancers (EBVaGC) tend to have a higher PD-L1 expression [47,48]. In a study by Kim et al. an unprecedented 100% response rate was observed in EBV positive metastatic gastric cancer patients [49]. Currently, a trial is evaluating the efficacy of Camrelizumab in stage 3 EBV+ MSI-H patients (ChiCTR1900027123) [50].

TMB can also be an important predictive biomarker. Kim et al. showed that the high-TMB advanced GC patients had a prolonged PFS after ICI treatment [51]. Similarly, another study by Wang et al. showed that high TMB can be a predictive biomarker for OS in advanced GC patients treated with toripalimab, an anti-PD1 antibody [52].

GEC is geographically unequally distributed. Gastric cancer has high incidence rates in Asia and Latin America [53]. This is related to a higher incidence of Helicobacter pylori in these areas [53]. However, studies have shown a difference in survival benefits between ethnic groups even within the United States [54]. Asians are known to have better survival than North Americans. A meta-analysis evaluating gene signatures in Asians vs. Non-Asians showed that the two groups exhibited different gene signatures [55]. All these factors could lead to variation in response to ICI therapy, as was seen in our study. GEC is heterogeneously distributed in terms of histology as well. Whereas ESCC is the most common sub-type worldwide, especially in East Asia, Africa, and Eastern Europe; esophageal adenocarcinoma (EAC) is the most common sub-type in the U.S. and Northern Europe [56]. EAC incidence has increased in Western countries in recent decades. It occurs predominantly in the lower esophagus near the GE junction and is associated with obesity, gastric reflux, and Barrett’s esophagus [57]. On the other hand, ESCC predominates in the upper and mid-esophagus and are associated with smoking and alcohol exposure in Western populations. A study by Kim et al. on esophageal cancers showed differences in the mutations based on histological subtypes. ESCC showed frequent genomic amplifications of *CCND1* and *SOX2*, and *TP63*, whereas *ERBB2*, *VEGFA* and *GATA4*, and *GATA6* were more commonly amplified in EAC [47]. In terms of molecular features, EAC resembled more to the gastric adenocarcinoma than ESCC. In our analysis, the esophageal location was found to have more benefit from ICIs than other locations. Histology-wise, squamous carcinoma showed more benefit to ICIs than adenocarcinoma. While genetic makeup could be responsible for these subgroup differences, additional translational studies are needed in this area [58].

## 6. Limitations

Our study has several strengths and limitations. To our knowledge, this is the largest meta-analysis investigating the role of ICIs in advanced GEC. A large number of patients allowed us to look at different settings and subgroups. On the other hand, we only included Phase 2/3 RCTs with survival data. While looking at the racial differences, only Asians and Whites were included. The non-Asian, non-White groups were not included, primarily due to inconsistent reporting. Second, to define the PD-L1 positive subgroup, we included patients with a PD-L1 CPS score ≥ 1; however, a subgroup analysis of Checkmate-648 study has shown that tumor cell PD-L1 expression, very similar to TPS, also has clinical utility in ESCC. Due to lack of consistent data, we could not perform analysis on other predictive biomarkers for ICI including TMB, STK-11, EBV. STK11 mutations are associated with primary resistance to PD-1/PD-L1 agents in non-small cell lung cancer; however, their role in metastatic gastric cancer has not been well established [59]. Due to lack of consistent data, we could not perform analysis on other predictive biomarkers for ICI including TMB, STK-11, EBV. Finally, some of the included studies have shorter follow-up, which may bias against long-term efficacy. Some subgroup-level data were not available for specific trials. Therefore, our results should be interpreted carefully.

## 7. Conclusions

This meta-analysis showed that although there was a general benefit of using ICIs in advanced GEC, not all patients benefit similarly. Both young and old patients respond similarly to anti-PD-1/PDL-1 ICIs, and more trials are needed in the older population. In PDL-1 negative patients, ICIs do not appear to add benefit and other treatment options should be investigated. Males, esophageal location, Asians, and squamous histology derive more benefit from ICI than females, gastric/gastroesophageal junction location, Whites, and adenocarcinoma histology, respectively. Further translational research is required to find the factors that influence ICI response.

## Figures and Tables

**Figure 1 pharmaceutics-14-01953-f001:**
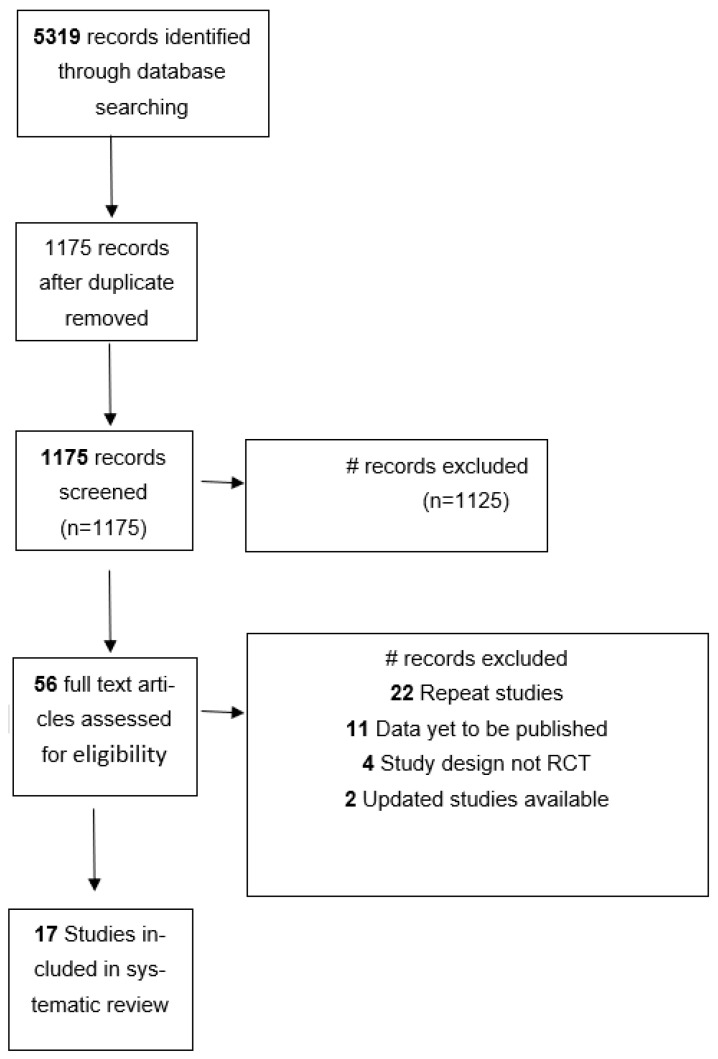
PRISMA diagram showing study selection flowchart.

**Figure 2 pharmaceutics-14-01953-f002:**
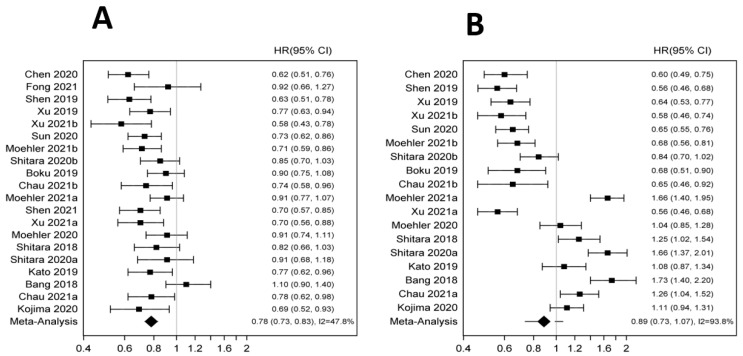
(**A**): Forest plot for overall survival in the total population. (**B**): Forest plot for progression free survival in the total population [5,6,13,14,15,16,17,18,19,20,21,22,23,24,25,26,27].

**Figure 3 pharmaceutics-14-01953-f003:**
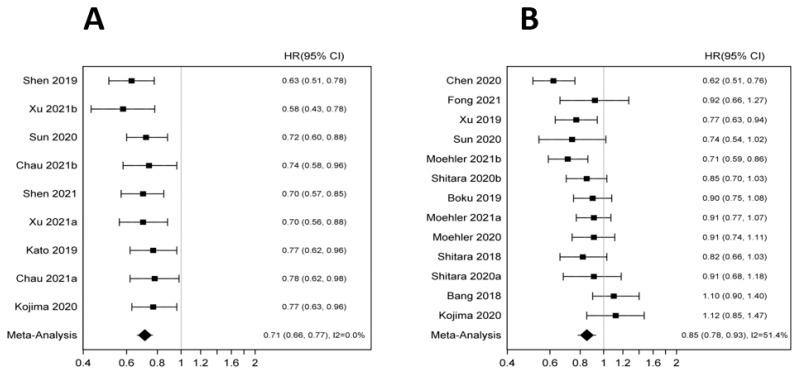
(**A**): Forest plot for overall survival in squamous cell carcinoma. (**B**): Forest plot for overall survival in adenocarcinoma [5,6,13,14,15,16,17,18,19,20,21,22,23,24,25,26,27].

**Figure 4 pharmaceutics-14-01953-f004:**
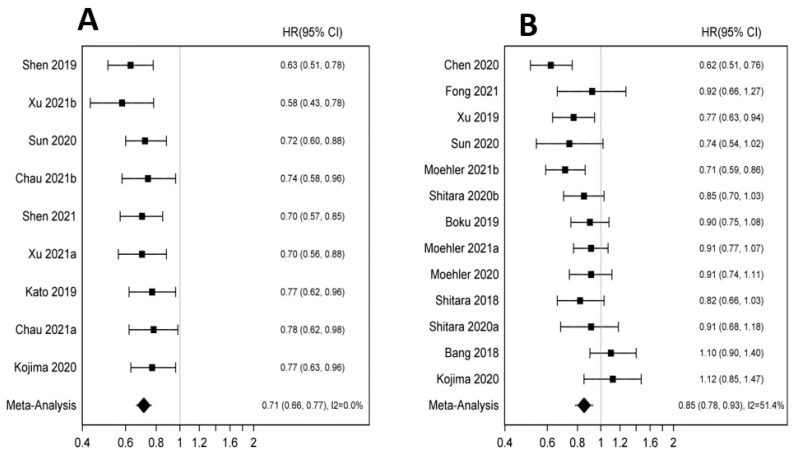
(**A**): Forest plot for overall survival in immunotherapy plus chemotherapy vs. chemotherapy. (**B**): Forest plot for progression free survival immunotherapy plus chemotherapy vs. chemotherapy [5,6,13,14,15,16,17,19,22].

**Table 1 pharmaceutics-14-01953-t001:** Included studies in the subgroup immunotherapy plus chemotherapy vs. chemotherapy.

		Patients Number		OS				PFS			>Grade 3 TRAE		TRAE	
				Median, mo				Median, mo						
Author	ICI	ICI	SoC	ICI	SoC	HR (95% CI)	P	ICI	SoC	HR (95% CI)	ICI	SoC	ICI	SoC
Chau et al., 2021 [19], CHECKMATE 648	Nivolumab + Chemotherapy	321	324	13.2	10.7	0.74 (0.58–0.96)	0.0021			0.65 (0.46–0.92)	47.00%	36.00%	96.00%	80.00%
Xu et al., 2021 [13], ESCORT	Camrelizumab + Chemotherapy	298	298	15.3	12	0.70 (0.56–0.88)	0.001	6.9	5.6	0.56 (0.46–0.68)	63.40%	67.70%	99.30%	97%
Boku et al., 2019 [14], ATTRACTION 4	Nivolumab + Chemotherapy	362	362	17.45	17.15	0.90 (0.75–1.08)	0.257	10.45	8.34	0.68 (0.51–0.90)	57.10%	48.60%	97.80%	97.50%
Shitara et al., 2020 [18], KEYNOTE 062	Pembrolizumab + Chemotherapy	257	250	12.5	11.1	0.85 (0.70–1.03)	0.05	6.9	6.4	0.84 (0.70–1.02)	73.20%	69.30%	94.00%	91.80%
Moehler et al., 2021 [5], CHECKMATE 649	Nivolumab + Chemotherapy	789	792	13.8	11.6	0.71 (0.59–0.86)	0.0002	7.7	6.9	0.68 (0.56–0.81)	59.00%	44.00%	95.00%	88.00%
Sun et al., 2020 [6], KEYNOTE 590	Pembrolizumab + Chemotherapy	373	376	12.6	9.8	0.73 (0.62–0.86)	<0.0001	6.3	5.8	0.65 (0.55–0.76)	86.00%	83.00%	100.00%	99.00%
Rui Hua Xu et al., 2021 [15], JUPITER-06	Toripalimab + Chemotherapy	257	257	17	11	0.58 (0.43–0.78)	0.00036	5.7	5.5	0.58 (0.46–0.74)	97.30%	56.00%	97.30%	64.60%
Jianming Xu et al., 2019 [16], ORIENT-16	Sintilimab + Chemotherapy	327	323	15.2	12.3	0.77 (0.63–0.94)	0.009	7.1	5.7	0.636 (0.525–0.771)	96.30%	52.50%	97.30%	59.80%
Lin Shen et al., 2019 [17], ORIENT-15	Sintilimab + Chemotherapy	327	332	16.7	12.5	0.628 (0.508–0.777)	<0.0001	7.2	5.7	0.558 (0.461–0.676)	98.20%	54.50%	98.20%	59.90%

Abbreviations: HR, hazard ratio, ICI, immune checkpoint inhibitor; OS, overall survival; PFS, profession-free survival; SoC, standard care; TRAE, treatment-related adverse event; mo, months.

**Table 2 pharmaceutics-14-01953-t002:** Included studies in the subgroup immunotherapy vs. chemotherapy.

		Patients Number		Os				PFS			>Grade 3 TRAE		TRAE	
				Median, mo				Median, mo						
Author	ICI	ICI	SoC	ICI	SoC	HR (95% CI)	P	ICI	SoC	HR (95% CI)	ICI	SoC	ICI	SoC
Kojima et. al., 2020 [22], KEYNOTE 181	Pembrolizumab	314	314	9.3	6.7	0.69 (0.52–0.93)	0.0074	2.1	3.4	1.11 (0.94–1.31)	18.20%	40.90%	64.30%	86.10%
Chau et al., 2021 [19], CHECKMATE 648	Nivolumab + Ipilimumab	325	324	12.8	10.7	0.78 (0.62–0.98)	0.011	2.9	5.6	1.26 (1.042–1.52)	32.00%	36.00%	90.00%	80.00%
Bang et. al., 2018 [20] JAVELIN 300	Avelumab	185	186	4.6	5	1.1 (0.9–1.4)	0.81	1.4	2.7	1.73 (1.4–2.2)	9.20%	31.60%	48.90%	74.00%
Kato et. al., 2019 [21], ATTRACTION 3	Nivolumab	210	209	10.9	8.4	0.77 (0.62–0.96)	0.02	1.7	3.4	1.08 (0.87–1.34)	18.00%	63.00%	66.00%	95%
Shitara et. al., 2020 [22], KEYNOTE 062	Pembrolizumab	256	250	10.6	11.1	0.91 (0.68–1.18)	NA	2	6.4	1.66 (1.37–2.01)	16.90%	69.30%	54.30%	91.80%
Shitara et. al., 2018 [25], KEYNOTE 061	Pembrolizumab	296	296	9.1	8.3	0.82 (0.66–1.03)	0.0421	1.5	4.1	1.25 (1.02–1.54)	14.00%	35.00%	53.00%	84.00%
Moehler et. al., 2020 [23], JAVALIN 100	Avelumab	249	250	10.4	10.9	0.91 (0.74–1.11)	0.1179	3.2	4.4	1.04 (0.85–1.28)	12.80%	32.80%	61.30%	77.30%
Shen et al., 2021 [24], RATIONALE 302	Tislelizumab	256	256	8.6	6.3	0.70 (0.57–0.85)	0.0001			NA	18.8.0%	55.8.0%	73.30%	93.80%
Moehler et. al., 2021 [5], CHECKMATE 649	Nivolumab + lpilimumab	409	404	11.7	11.8	0.91 (0.77–1.07)	NA	2.8	7.1	1.66 (1.40–1.95)	38.00%	46.00%	80.00%	92.00%

Abbreviations: HR, hazard ratio, ICI, immune checkpoint inhibitor; OS, overall survival; PFS, profession-free survival; SoC, standard of care; TRAE, treatment-related adverse event; mo, months.

## Data Availability

The data presented in this study are openly available in Appendix A.

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
