# Peer review of "Anti PD-1/Anti PDL-1 Inhibitors in Advanced Gastroesophageal Cancers: A Systematic Review and Meta-Analysis of Phase 2/3 Randomized Controlled Trials"

_pharmaceutics, 2022, doi:10.3390/pharmaceutics14091953_

Round 1

Reviewer 1 Report

Manuscript entitled "Anti PD-1 /Anti PDL-1 Inhibitors in Advanced Gastroesophageal Cancers: A Systematic Review and Meta-Analysis of Phase 2/3 Randomized Controlled Trials"

This work is potentially interesting but some modifications should be made before it can be published.

Major issues:

1. This work lacks an important element, that is "biomarker study". The biomarker expression should be presented and discussed for PD-1, PD-L1, TMB, .... . 

2. The predictive markers such as STK11 in post-Hoc analysis should also be included.

Author Response

Reviewer 1

  1. This work lacks an important element, that is "biomarker study". The biomarker expression should be presented and discussed for PD-1, PD-L1, TMB, .... .

Answer: We appreciate the reviewer's comment on biomarker analysis. The biomarker section has been further expanded in the study as below:

“Several predictive biomarkers have been identified in addition to PD-L1 expression, including MSI, tumor mutational burden (TMB), and EBER (PMID: 31056702). A post-hoc analysis of the KEYNOTE-059, KEYNOTE-061 and KEYNOTE-062 showed that the MSI-H gastric/GEJ cancer patients derived benefit from Pembrolizumab, regardless of the line of therapy in which it was received (PMID: 33792646). A recent study showed that nearly 9% of GC patients are Epstein-Barr Virus (EBV) positive (Murphy, G., Pfeiffer, R., Camargo, M. C. & Rabkin, C. S. Gastroenterology 137, 824–833 (2009)), and EBV-associated gastric cancers (EBVaGC) tend to have a higher PD-L1 expression (doi: 10.1007/s12672-022-00479-0). In a study by Kim et al. an unprecedented 100% response rate was observed in EBV positive metastatic gastric cancer patients (Kim ST, Cristescu R, Bass AJ, et al. Comprehensive molecular characterization of clinical responses to PD-1 inhibition in metastatic gastric cancer. Nat Med. 2018;24(9):1449-1458. doi:10.1038/s41591-018-0101-z) Currently, a trial is evaluating the efficacy of Camrelizumab in stage 3 EBV+ MSI-H patients (ChiCTR1900027123).

TMB can also be an important predictive biomarker. Kim et al. showed that the high-TMB advanced GC patients had a prolonged PFS after ICI treatment. (DOI: 10.3389/fonc.2020.00314). Similarly, another study by Wang et al showed that high TMB can be a predictive biomarker for OS in advanced GC patients treated with toripalimab, an anti-PD1 antibody (Wang F, Wei XL, Wang FH, et al. Safety, efficacy and tumor mutational burden as a biomarker of overall survival benefit in chemo-refractory gastric cancer treated with toripalimab, a PD-1 antibody in phase Ib/II clinical trial NCT02915432. Ann Oncol. 2019;30(9):1479-1486. doi:10.1093/annonc/mdz197)”.

  1. 2. The predictive markers such as STK11 in the posthoc analysis should also be included.

Ans: Again, we would like to thank the reviewer for this comment. We could not perform a post hoc analysis due to the lack of data in the RCT about STK-11. This was added under our limitations:

“Due to lack of consistent data, we could not perform analysis on other predictive biomarkers for ICI including TMB, STK-11, EBV.”

We have also added the line below on this important marker in our Discussion section:

“STK11 mutations are associated with primary resistance to PD-1/PD-L1 agents in non-small cell lung cancer; however, their role in metastatic gastric cancer has not been well established.”

Reviewer 2 Report

I read with great interest this paper based on a meta-analysis on phase 2/3 RCT of anti PD-1/PD-L1 inhibitors in advanced gastroesophageal cancers. The study is well-designed and the results are of interest for the scientific community facing with this topic. There are some suggestions for improving this manuscript:

1) please add representative histological and immunohistochemical images (microscropic photograph) of tumor tissue and PD-1/PD-L1 IHC. This is very useful for the readers.

2) please discuss (more) in depth the role of IHC in the assessment of PD-L1 and of the different clones and platforms in clinical practice: issues and advantages.

3) please discuss the role of immunotherapy in GI district based not only on PD-1/PD-L1 but also on the others important markers in this setting, such as TMB and MSI, highlighting similarities and differences ( suggested refererences: PMID: 31056702 ; 35694998 )

4) please indicate heterogeneity of the results of meta-analysis (I2) and add this index also in the figures. The figures should have a better resultion, too.

Author Response

Reviewer 2

  • please add representative histological and immunohistochemical images (microscropic photograph) of tumor tissue and PD-1/PD-L1 IHC. This is very useful for the readers.

Ans: We appreciate the reviewer's comment. Since this study was not done on an individual patient level, we, unfortunately, do not have slides/microscopic images of the tumor tissue.

  • Please discuss (more) in depth the role of IHC in the assessment of PD-L1 and the different clones and platforms in clinical practice: issues and advantages.

Ans: We agree with more details about IHC. We have added the following lines in the discussion section:

“The tumor proportion score (TPS) score is the ratio of the number of PD-L1–expressing tumor cells to all tumor cells. The combined positive score (CPS) was developed to consider the expression of PD-L1 on tumor and immune cells combined. CPS is the ratio of all PD-L1–expressing cells (tumor cells, lymphocytes, macrophages) to the number of all tumor cells. In gastric cancer, CPS has been found to be a better biomarker than TPS [Yamashita, K., Iwatsuki, M., Harada, K. et al. Gastric Cancer 23, 95–104 (2020). https://doi.org/10.1007/s10120-019-00999-9]. Measurement of PD-L1 expression with the CPS in patients in KEYNOTE-059 trial showed that the objective response to pembrolizumab was significantly associated with CPS but not with TPS (DOI: 10.1200/JCO.2018.36.15_suppl.4065 Journal of Clinical Oncology 36, no. 15_suppl (May 20, 2018) 4065-4065). However, there is a discrepancy in the assessment methods as they require different staining protocols, equipment, and cut-offs, and standardized methods and definitions of PDL-1 positivity are required. In a recent study, Yeong et al showed that Dako 28–8 assay results in higher PD-L1 CPS positives compared to the Dako 22C3 assay. Therefore, these assays should not be used interchangeably while assessing the PD-L1 CPS scores in patients.”

  • Please discuss the role of immunotherapy in GI district based not only on PD-1/PD-L1 but also on the others important markers in this setting, such as TMB and MSI, highlighting similarities and differences (suggested references: PMID: 31056702; 35694998)

Ans: please note that we have added the following to the discussion section:

“Several predictive biomarkers have been identified in addition to PD-L1 expression, including MSI, tumor mutational burden (TMB), and EBER (PMID: 31056702). A post-hoc analysis of the KEYNOTE-059, KEYNOTE-061 and KEYNOTE-062 showed that the MSI-H gastric/GEJ cancer patients derived benefit from Pembrolizumab, regardless of the line of therapy in which it was received (PMID: 33792646). A recent study showed that nearly 9% of GC patients are Epstein-Barr Virus (EBV) positive (Murphy, G., Pfeiffer, R., Camargo, M. C. & Rabkin, C. S. Gastroenterology 137, 824–833 (2009)), and EBV-associated gastric cancers (EBVaGC) tend to have a higher PD-L1 expression (doi: 10.1007/s12672-022-00479-0). In a study by Kim et al. an unprecedented 100% response rate was observed in EBV positive metastatic gastric cancer patients (Kim ST, Cristescu R, Bass AJ, et al. Comprehensive molecular characterization of clinical responses to PD-1 inhibition in metastatic gastric cancer. Nat Med. 2018;24(9):1449-1458. doi:10.1038/s41591-018-0101-z) Currently, a trial is evaluating the efficacy of Camrelizumab in stage 3 EBV+ MSI-H patients (ChiCTR1900027123).

TMB can also be an important predictive biomarker. Kim et al. showed that the high-TMB advanced GC patients had a prolonged PFS after ICI treatment. (DOI: 10.3389/fonc.2020.00314). Similarly, another study by Wang et al showed that high TMB can be a predictive biomarker for OS in advanced GC patients treated with toripalimab, an anti-PD1 antibody (Wang F, Wei XL, Wang FH, et al. Safety, efficacy and tumor mutational burden as a biomarker of overall survival benefit in chemo-refractory gastric cancer treated with toripalimab, a PD-1 antibody in phase Ib/II clinical trial NCT02915432. Ann Oncol. 2019;30(9):1479-1486. doi:10.1093/annonc/mdz197)”.

  • Please indicate heterogeneity of the results of meta-analysis (I2) and add this index also in the figures. The figures should have a better resolution, too.

We have added I2 to all our images. We have redone all the images on Power-point to increase resolution. 

Round 2

Reviewer 1 Report

This work is acceptable for publication in the present form.